# Neutrophil as a Carrier for Cancer Nanotherapeutics: A Comparative Study of Liposome, PLGA, and Magnetic Nanoparticles Delivery to Tumors

**DOI:** 10.3390/ph16111564

**Published:** 2023-11-06

**Authors:** Anastasiia S. Garanina, Daniil A. Vishnevskiy, Anastasia A. Chernysheva, Marat P. Valikhov, Julia A. Malinovskaya, Polina A. Lazareva, Alevtina S. Semkina, Maxim A. Abakumov, Victor A. Naumenko

**Affiliations:** 1Laboratory of Biomedical Nanomaterials, National University of Science and Technology «MISIS», 119049 Moscow, Russia; abakumov1988@gmail.com; 2Department of Medical Nanobiotechnology, N.I. Pirogov Russian National Research Medical University, 117997 Moscow, Russia; vishnevskiy.daniil.andreevich@gmail.com (D.A.V.); marat.valikhov@gmail.com (M.P.V.); polizara604@gmail.com (P.A.L.); alevtina.semkina@gmail.com (A.S.S.); 3V. Serbsky National Medical Research Center for Psychiatry and Narcology, 119034 Moscow, Russia; aachernysheva512@gmail.com (A.A.C.); naumenko.vict@gmail.com (V.A.N.); 4D. Mendeleev University of Chemical Technology of Russia, 125047 Moscow, Russia; j.malinowskaya@gmail.com

**Keywords:** nanoparticles, neutrophils, cell-based delivery system, tumor, intravital microscopy

## Abstract

Insufficient drug accumulation in tumors is still a major concern for using cancer nanotherapeutics. Here, the neutrophil-based delivery of three nanoparticle types—liposomes, PLGA, and magnetite nanoparticles—was assessed both in vitro and in vivo. Confocal microscopy and a flow cytometry analysis demonstrated that all the studied nanoparticles interacted with neutrophils from the peripheral blood of mice with 4T1 mammary adenocarcinoma without a significant impact on neutrophil viability or activation state. Intravital microscopy of the tumor microenvironment showed that the neutrophils did not engulf the liposomes after intravenous administration, but facilitated nanoparticle extravasation in tumors through micro- and macroleakages. PLGA accumulated along the vessel walls in the form of local clusters. Later, PLGA nanoparticle-loaded neutrophils were found to cross the vascular barrier and migrate towards the tumor core. The magnetite nanoparticles extravasated in tumors both via spontaneous macroleakages and on neutrophils. Overall, the specific type of nanoparticles largely determined their behavior in blood vessels and their neutrophil-mediated delivery to the tumor. Since neutrophils are the first to migrate to the site of inflammation, they can increase nanodrug delivery effectiveness for nanomedicine application.

## 1. Introduction

Nanomedicine for cancer treatment is an evolving field of science [1]. Nanoparticles (NPs) can be used in diagnostics [2], as well as in the targeted delivery of drugs to tumor sites, reducing their toxicity to the whole organism [3]. However, the problem of nanodrug accumulation efficiency in tumor tissues is still relevant for nanomedicine. One of the proposed approaches to overcoming this obstacle is the use of nanodrug cell-based delivery systems (NCBDs) [4,5,6], which are currently under active investigation. NCBDs can be prepared using two main strategies: (1) the ex vivo loading of cells with NPs and subsequent injection of the established NCBDs back into the body; or (2) the injection of NPs into an organism with the subsequent uptake of NPs by specific cells in vivo (e.g., phagocytes) that can deliver NPs to tumor lesions. Several types of cells have been proposed as drug delivery systems, including mesenchymal stem cells, leukocytes, and red blood cells [4]. The most promising of these cells are peripheral blood neutrophils. This is due to their large numbers (65–80% of circulating leukocytes), the ability to cross the vascular barrier and to migrate over long distances in the body in response to the chemokines produced in sites of inflammation [7] and tumor lesions. However, the success of using neutrophils as NCBDs largely depends on the efficiency of the cell loading with NPs. There are many studies describing the interaction of neutrophils with NPs [8,9,10,11]. The effect of NPs on neutrophils, in particular, their toxicity and ability to alter cell function, was investigated by Liz et al. The authors demonstrated that 15 and 20 nm silver NPs were internalized by neutrophils and led to cell death in a dose-dependent manner, while 70 nm particles were not captured by the cells and had a cytoprotective effect [12]. Moreover, Skallberg et al. found that 15 nm silver NPs caused an increase in IL-1β secretion, the hyperproduction of reactive oxygen species (ROS), and the formation of neutrophil extracellular traps (NETs) [13]. Bisso et al. studied the response of neutrophils to NPs with different sizes and chemistries [14]. Lipid, poly(styrene), poly (lactic-co-glycolic acid) (PLGA), and gold NPs ranging in diameter from 5 nm to 2 μm were investigated. It was shown that neutrophils preferentially internalized larger NPs up to 200 nm. NPs of all the chemical compositions analyzed had neither a stimulating nor an inhibitory effect on neutrophil viability and function. However, Babin et al. revealed that TiO_2_, CeO_2_, or ZnO NPs caused neutrophil activation [15]. On the contrary, Verdon et al. did not find any significant effect of TiO_2_ NPs on the metabolic activity of primary neutrophils or the release of NETs by the cells [16]. The other characteristics of NPs that may influence their interaction with neutrophils and the function of the latter are charge and surface modification. It has been shown that the surface modification of liposomes and PLGA NPs with CTAB (hexadecyltrimethyl-ammonium bromide) or SME (soyaethyl morpholinium ethosulfate) prevents their internalization by neutrophils [17]. Furthermore, the surface modification of various cationic NPs (nanobubbles, liposomes, and PLGA NPs) with CTAB or SME resulted in human neutrophil membrane damage, accompanied by the release of lactate dehydrogenase (LDH) and neutrophil elastase, increased intracellular Ca^2+^ levels, and the formation of ROS [17,18,19]. When comparing the data on the effect of NPs on neutrophils, it is difficult to identify a single parameter that plays a critical role in cytotoxicity/cytoprotection or the activation/inhibition of the cells.

Studies of drug-loaded NPs have recently recognized the efficacy of neutrophil-mediated delivery [20]. Chu et al. showed a reduction in tumor growth and an increase in animal survival after combined immunotherapy and intravenous (i.v.) administration of albumin NPs loaded with photosensitizers, as well as after the neutrophil-mediated delivery of gold nanorods modified with antibodies against CD11b and following photosensitivity [8,21]. Xue et al. demonstrated the suppression of postoperative malignant glioma through the neutrophil delivery of paclitaxel-loaded liposomes after i.v. injection of NPs [22]. Drug-loaded liposomes with polysialic-acid-modified surfaces and PLGA NPs were found to be taken up by neutrophils and transferred to the tumor site after systemic administration [23,24]. Hao et al. reported that the combination of paclitaxel-PLGA NPs with the pre-implantation of CXCL1-loaded hydrogels at the tumor site resulted in a better tumor suppression than that with paclitaxel-PLGA NPs alone or paclitaxel-liposomes [24]. The authors explained this through the continuous release of CXCL1 from the hydrogel, which attracted NP-loaded neutrophils and concentrated them at the implanted site. Li et al. also demonstrated the uptake of doxorubicin-loaded liposomes modified with sialic acid conjugate by peripheral blood neutrophils [25]. The resulting neutrophil-mediated drug delivery system was shown to be effective in the treatment of murine sarcoma S180. Shen et al. demonstrated the efficiency of neutrophil-based doxorubicin-loaded liposomal targeted therapy for residual tumors after high-intensity focused ultrasound ablation [26]. This approach allowed targeted drug delivery through ablation-induced inflammation and reduced systemic chemotherapy side effects. The effectiveness of ex vivo cell loading with a therapeutic agent to create NCBDs for tumor drug delivery has been demonstrated in mouse, rat, and human glioma models [22,27]. However, the migration of neutrophils to the tumor was ensured by post-operative inflammation. Thus, there are two main strategies for enhancing the targeting ability of neutrophils to improve the efficacy of antitumor therapy: (1) the induction of the inflammatory microenvironment in tumors via surgery, irradiation, and photothermal therapy to facilitate the recruitment of chemokines; and (2) the functionalization of NPs with targeting by surface modifications [28,29]. However, modifications of NPs may affect neutrophil biology. For example, there is a possibility that coating the surfaces of NPs with antigen against CD11b may lead to impaired neutrophil migration [21]. Fromen et al. also showed that the interaction of neutrophils with polysterol NPs containing carboxyl groups leads to a decrease in cell adhesion function and their migration to the liver. This factor renders such NPs unsuitable for tumor drug delivery [30]. Other challenges for the use of neutrophils as drug delivery vehicles relate to the ex vivo approach for cell loading with NPs [11]. The short lifespan (half-life of 1–5 days) of neutrophils necessitates the isolation, purification, drug loading, and reinfusion of NP-loaded cells within a few hours. Finally, the rapid intracellular degradation of the loaded cargo, high cost, and insufficient volume of harvested cells limit the efficacy of NCBDs. Thus, despite the fact that the neutrophil-mediated delivery of anticancer nanodrugs shows promising results in the fight against tumors, many aspects of the interaction of NPs with neutrophils, especially in the bloodstream after i.v. injection, remain unclear. In addition, to the best of our knowledge, comparative studies of the behaviors of different NP types in the tumor microenvironment in vivo are lacking.

There is growing interest in the prospects of the neutrophil-based delivery of liposomes, polymeric NPs, and magnetic NPs, which have been approved for cancer treatment [11,31]. Liposomes demonstrate a low systemic toxicity, biodegradability, and the ability to carry large amounts of drugs and protect them from physiological degradation, thereby prolonging their half-life [32]. PLGA, a synthetic polymer, is now widely used for the preparation of NPs [11]. It is flexible, readily biodegradable, and allows for the release of different cargoes to be regulated. Magnetic NPs are attractive as a theranostic tool, enabling both drug delivery and the non-invasive visualization of tumor lesions.

In this study, we first investigated the effect of three NP types with different physicochemical properties (liposomes, PLGA, and magnetite nanoparticles) on neutrophils both in vitro and in vivo. All the NPs were shown to be non-toxic to neutrophils, but differed slightly in their effects on cells isolated from tumor-bearing mice. Intravital microscopy revealed the difference between NP types in their microdistribution in the tumors and interactions with neutrophils. Magnetite NPs extravasated mainly via spontaneous macroleakages, but could also be internalized by neutrophils and delivered to the tumor. PLGA NPs accumulated along the vessel wall in the form of local clusters. This type of NPs was also taken up by neutrophils migrating to the tumor site. Liposomes were never captured by neutrophils; however, neutrophils were able to initiate liposome delivery to tumors through micro- and macroleakages. Thus, real-time in vivo observations clearly demonstrated that the interaction of neutrophils with NPs depends on the specific type of the latter. Further studies on the mechanisms of NPs’ interaction with neutrophils will help to design NPs with improved tumor targeting.

## 2. Results

### 2.1. Nanoparticle Characterization

Three types of nanoparticles (NPs) with different physicochemical parameters were analyzed: (1) magnetite NPs (MNPs) coated with human serum albumin; (2) NPs based on copolymers of lactic and glycolic acids (PLGA); and (3) PEGylated liposomes. An examination of the NPs with transmission electron microscopy revealed that they were all spherical in shape (Figure 1). The study of the NPs’ size and ζ-potential using the dynamic light scattering (DLS) method revealed that the MNPs had a hydrodynamic size of 35 nm and a ζ-potential of (−26 ± 2) mV, PLGA NPs—hydrodynamic size of 100–120 nm and a ζ-potential of (−23 ± 9) mV, and liposomes—hydrodynamic size of about 120 nm and a ζ-potential of (−2.6 ± 19.4) mV. All the NPs were modified with the following fluorescent dyes: MNPs and PLGA NPs—with Cyanine 5 amine (Cy5), and liposomes—with 1,1′-dioctadecyl-3,3,3′,3′-tetramethylindodicarbocyanine, 4-chlorobenzenesulfonate (DiD).

### 2.2. Nanoparticles Interaction with and Effect on Neutrophils

The 4T1 (murine mammary adenocarcinoma) tumor model was studied because it causes an increase in the number of neutrophils in the blood (Appendix A).

The in vitro study of the interaction of the NPs with neutrophils blood isolated from the tumor-bearing mice showed a gradual increase in NP accumulation with the time of co-incubation (Figure 2a–c). The PLGA-Cy5 NPs’ accumulation reached a plateau after 30 min of incubation with cells and did not change significantly thereafter (Figure 2b). The accumulation of liposomes-DiD in cells was not statistically different after 1 and 2 h of co-incubation (Figure 2c). All types of NPs investigated were localized inside neutrophils in the form of conglomerates with a size of about 500 nm (Appendix A).

The in vivo study showed that less than 0.5% of neutrophils interacted with all types of NPs (Figure 2d–f). The accumulation of MNPs-Cy5 in cells did not change with time; the maximum of PLGA-Cy5 interaction with neutrophils was detected after 10 min of i.v. NP administration and then decreased; the peak of liposomes-DiD interaction with cells was reached after 1.5 h of NP injection. Almost the same results were obtained for other subpopulations of blood leukocytes: CD4^+^/CD8^+^ T-cells and B-cells. The most effective interaction of all three types of NPs with monocytes was shown: up to 15% of the subpopulation for MNPs-Cy5 and 80% for PLGA-Cy5 and liposomes-DiD. It should be noted that the accumulation of MNPs-Cy5 and PLGA-Cy5 in monocytes decreased over time, while the interaction of liposomes-DiD with these cells increased. Based on the results obtained, it can be concluded that monocytes are the most promising candidates for drug-loaded NP delivery to tumors. However, it should be taken into account that the number of monocytes in the blood is significantly lower compared to neutrophils (about 30 times) (Appendix A). Furthermore, the percentage of neutrophils in the 4T1 tumor microenvironment is quite high (32%) (Appendix A). Thus, in addition to the fact that neutrophils are the first to migrate to the site of inflammation [7], their quantitative predominance in the blood, and quite high number in the tumor microenvironment, as well as their ability to interact with the NPs studied both in vitro and in vivo, make these cells the preferred carriers for NP delivery to tumors.

One of the most important factors in the biomedical application of NPs is their toxicity to cells. In this regard, an LDH assay was performed for the NPs studied to evaluate their effect on the viability of the neutrophils isolated from the blood of 4T1-bearing mice. As a result, it was shown that none of the three types of NPs caused increased cell death, even at concentrations four times higher than those used in all the experiments described in this study (Figure 2g–i).

Another factor that determines the suitability of using NPs for neutrophil-based drug delivery to tumors is their effect on cell activation. The main characteristic signs of neutrophil activation are an increased production of reactive oxygen species (ROS) and expression of the surface marker CD11b [33]. The systemic administration of the NPs studied showed that the level of ROS production by the neutrophils that had taken up the NPs was higher (approximately 2-fold) compared to the control cells and cells that had not captured NPs (Figure 2j and Appendix A). However, a statistically significant increase in ROS generation was only found in the cells that captured liposomes-DiD. Nevertheless, the incubation of neutrophils with phorbol myristate acetate (PMA, a known neutrophil activator) resulted in a marked increase in ROS production by the cells (approximately 4.7-fold). Similar results were obtained for the marker CD11b. The level of its expression in the neutrophils that did not take up NPs was identical to that in the control cells, whereas, in the neutrophils that took up NPs, it was increased (approximately 2-fold) (Figure 2k and Appendix A). A statistically significant difference from the control group was found for the cells that interacted with PLGA-Cy5 NPs and liposomes-DiD. However, the expression of CD11b by the neutrophils incubated in the presence of PMA was significantly increased (approximately 12-fold) compared to all other experimental groups.

Thus, all the NPs analyzed in this study showed no toxicity and a minimal activation effect on neutrophils.

### 2.3. Nanoparticles Extravasation from Tumor Vessels

The behavior of NPs in the tumor microenvironment after systemic administration, their interaction with leukocytes, and their routes of extravasation were investigated using intravital microscopy (IVM). Small MNPs-Cy5 were shown to rapidly extravasate from blood vessels (within 8–10 min), mainly via spontaneous macroleakages penetrating deep into the tissue (Figure 3), and to be taken up by various cells of the tumor microenvironment, wherein neutrophils were found to capture MNPs-Cy5 in the bloodstream and extravasate with them (Appendix A). In addition, monocytes with MNPs-Cy5 conglomerates inside were observed to migrate through the vessel wall (Appendix A).

PLGA-Cy5 NPs extravasated by accumulating along the periphery of the vessels in the form of local clusters (of varying size) over 20–30 min after their systemic administration (Figure 4a–d). NPs were found to be taken up by neutrophils in the bloodstream (Figure 4e) and transported by neutrophils migrating to the tumor core (Figure 4f). To further investigate the localization of NP clusters, endothelial cells were stained with anti-CD31 antibodies. This showed that, despite the size of the clusters, they were located outside the vessels but in close proximity to the endothelial cells (Appendix A), wherein PLGA-Cy5 clusters could be pinpointed, elongated along the vessel wall or “hugging” the vessel. Based on the morphology of the clusters, we hypothesized that NPs could interact with pericytes. To test this hypothesis, tumors were stained with NG2 (neuron-glial antigen 2) and α-SMA (α-smooth muscle actin) antibodies—the main markers of pericytes [34]. It was found that the fluorescent signal from the PLGA-Cy5 NPs could co-localize with that from pericytes (Appendix A) or not (Appendix A). Thus, PLGA-Cy5 cluster formation can be associated with NP accumulation in pericytes, as well being independent of these cells. Finally, similar to magnetic NPs, PLGA-Cy5 were shown to be taken up by monocytes in the blood vessels, as well as in the tumor microenvironment (Appendix A).

The liposomes-DiD extravasated due to the formation of microleakages (local perivascular NP deposition) (Figure 5a–c,e–h) and macroleakages (extensive diffusion penetrating deep into the tumor tissues) (Figure 5d). The leakages could be mediated by neutrophils or occur independently of these cells [35]. The process of neutrophil-associated liposome-DiD microleakage formation is shown in Figure 5e–h. It can be seen that neutrophil attaches to the vessel wall, passes through it, and migrates into the surrounding tissue. At the same time, its morphology changes. The initially rounded shape of the circulating neutrophil becomes irregular, a pseudopod appears at the leading edge of the migrating cell, and there is thin plasma membrane formation at the rear edge. In parallel, liposomes-DiD exit the vessel following the extravasated neutrophil, resulting in the formation of a microleakage. Over time, NPs accumulate in this leakage, as indicated by an increase in the fluorescence signal of the DiD dye. It should be noted that liposomes-DiD are not taken up by neutrophil during this process. Cases of macroleakage were rare. Similar to the PLGA-Cy5 NPs, the investigation of liposome-DiD interaction with pericytes showed that NP microleakage may be associated with or independent of these cells (Appendix A). Previously, we also demonstrated that liposomes-DiD interact with monocytes, however, the latter do not penetrate through the vessel wall in this case [35].

To confirm the different routes of the NPs’ extravasation, two types of NPs were injected simultaneously—MNPs-Cy3 and liposomes-DiD. As a result, a mismatch of leakage sites was detected—MNPs-Cy3 were absent in the regions where liposomes-DiD microleakages were formed (Figure 6). At the same time, magnetite NPs were visualized inside the cells of the tumor microenvironment, accumulated by passive transport, while liposomes-DiD were not found there. Thus, it was concluded that the loci of spontaneous MNPs-Cy3 leakages and liposomes-DiD microleakages were not related.

Finally, the NPs differed in terms of circulation time. It was observed that the complete clearance of PLGA-Cy5 from the vessels occurred 40 min after administration, whereas MNPs-Cy5 and liposomes-DiD circulated for a longer time (Appendix A). The NPs with the longest circulation time were liposomes-DiD (more than 1 h).

Thus, we demonstrated that all types of NPs investigated did not have a significant impact on neutrophils and can be used for neutrophil-based drug delivery to tumors. However, the behavior of NPs in the blood vessels, the nature of their interaction with neutrophils, and, consequently, the contribution of neutrophil-based transport depend on the specific type of NPs.

## 3. Discussion

In this current study, three types of NPs were investigated to identify the most promising candidates for application in nanomedicine as carriers for neutrophil-based drug delivery to tumors. These NPs (liposomes, PLGA, and magnetic NPs) are the most commonly used NPs for tumor diagnosis and therapy [36,37,38]. In addition, liposomes have properties similar to those of clinically used NPs, such as Doxil, which has shown survival benefits in patients with ovarian cancer [39].

The NPs were compared according to the following criteria: (1) toxicity to neutrophils; (2) interaction with and accumulation in neutrophils; (3) effect on neutrophil activation (increased ROS production and CD11b expression); (4) behavior in the blood vessels of the tumor microenvironment after systemic administration; and (5) routes of extravasation (neutrophil-mediated and non-neutrophil-related).

Consistent with previous reports, the PLGA-Cy5 and liposomes-DiD showed no neutrophil toxicity [14]. Similarly, MNPs-Cy5 coated with human serum albumin (HSA) did not induce the activation of cell death. All types of NPs accumulated in isolated neutrophils. Similar to the results of Che et al., a co-incubation time of 1 h was sufficient for cell loading with NPs [40]. It should be noted that the NPs with the largest diameter (120 nm liposomes-DiD) accumulated in the neutrophils more effectively than the smaller MNPs-Cy5 and PLGA-Cy5. The same data were obtained by Bisso et al. [14] and Hao et al. [24], who showed that neutrophils preferentially take up NPs with a size of 200–260 nm. In vivo studies also showed that NPs interact with neutrophils. The percentage of NP-laden neutrophils was not pronounced compared to monocytes, but due to the high abundance of neutrophils in the peripheral blood and their biological properties [7], these cells are still considered to be a promising platform for NCBDs [6]. Interestingly, the interaction of the liposomes-DiD with neutrophils differed depending on the experimental conditions, in vitro and in vivo. Liposomes accumulated inside the cells during in vitro experiments, similar to that shown by Che et al. [40]. However, the internalization of this type of NPs by neutrophils was not detected after i.v. injection [35]. This fact should be taken into account when modelling the experiments.

The effect of NPs on neutrophils is very important for understanding the physiological and pathophysiological consequences of NPs’ application [41,42,43]. The NPs studied were shown to induce a 2-fold increase in neutrophil ROS production and CD11b expression after their systemic administration. However, this result was statistically significant only for liposomes-DiD. The observed response may be due to the stimulatory effects of PEG on innate immune cells associated with allergy [44]. Nevertheless, the observed increases in these two indicators were insignificant compared to the effect of PMA. By studying the impact of different NPs (liposomes, silica, iron oxide, and functionalized single-walled carbon nanotubes) on leukocyte subpopulations in human whole blood ex vivo, Kermanizadeh et al. found that only carbon nanotubes caused a concentration-dependent and significant increase in ROS production in monocytes and neutrophils [45]. Liposomes and iron oxide NPs at concentrations up to 200 μg/mL did not result in pronounced changes in ROS levels compared to control cells. Freitas et al. found that ROS production by neutrophils depends on the size of the NPs. The authors showed that an oxidative burst of neutrophils was only induced by small AgNPs with sizes of 5 and 10 nm in a concentration-dependent manner, while 50 nm AgNPs had no effect on the cells [46]. Snoderly et al. reported that the effect of NPs on ROS production depends on the chemistry and composition of the NPs [47]. It was shown that only nano-encapsulated manganese oxide particles, but not nano-encapsulated iron oxide particles, caused increased ROS production by neutrophils. Finally, Bisso et al. showed that poly(styrene) and liposomal particles did not enhance human neutrophil activation [14]. Thus, the currently available information and the fact that the identified effects of the investigated NPs on neutrophils were insignificant imply the prospects of these NPs’ use as NCBDs.

NADPH oxidase is responsible for the ability of neutrophils to produce ROS [48]. Neutrophil ROS contribute to microbial killing, trigger the formation of neutrophil extracellular traps (NETs), and appear to be involved in inflammation control. NETs can be of both nuclear and mitochondrial origin [49]. Despite the current confusion in terminology, not all pathways of NET formation result in cell death. Therefore, Boeltz et al. suggested that the term “NETosis” (implying a specific form of programmed cell death) should be avoided or used only in contexts where the demise of neutrophils is obvious. In all other cases, it was recommended to use the term “NET formation” instead. PMA is a known neutrophil activator that induces ROS production by these cells [50]. In this study, we showed the significant differences in the levels of ROS production and CD11b expression by neutrophils incubated in the presence of PMA or interacting with three types of NPs. Both indicators were lower in the latter case. This fact allows us to conclude a lack of neutrophil activation by the NPs studied and a toxicity of the NPs to the cells.

Thus, the three types of NPs with different physicochemical parameters observed in this study had a similar effect on the neutrophils during the in vitro and in vivo investigations. However, they showed completely different behavior in the tumor microenvironment after systemic administration. MNPs-Cy5 rapidly extravasated due to spontaneous macroleakages and penetrated the tumor. PLGA-Cy5 formed clusters along the vessel wall and accumulated there. Both types of NPs could be taken up by neutrophils and transported into the tumor tissue. In contrast, the liposomes-DiD were never internalized by neutrophils and extravasated due to micro- or macroleakages. However, neutrophil migration across the vascular endothelium could induce the formation of such leakages [35]. The PLGA-Cy5 clusters and liposome-DiD microleakages were morphologically similar. Moreover, in some cases, both accumulated inside pericytes. However, in a large sample of animals, we did not observe the direct involvement of neutrophils in the formation of PLGA-Cy5 clusters, in contrast to liposome microleakages. Finally, it remains unclear what these pericyte-independent clusters and leakages are. We can propose that both PLGA-Cy5 and liposomes-DiD can accumulate under the endothelial basement membrane and persist in the perivascular space [35]. Previously, it has been shown that magnetic NPs with a size of about 100 nm, similar to the PLGA-Cy5 NPs studied, exhibit the same behavior as PLGA NPs in vivo [51]. Thus, the described clusters/microleakages may be a characteristic extravasation method for NPs with size of ~100 nm, independent of NP chemistry. A further detailed investigation of 100 nm MNP microleakages using transmission electron microscopy with a higher resolution compared to IVM can shed light on the morphology of this formation.

Studies describing neutrophil-mediated NPs’ delivery using intravital microscopy are limited. They mainly evaluate the efficacy of this type of tumor therapy. Existing studies using intravital imaging describe negatively charged NPs with a modified surface and a size greater than 100 nm that interact with neutrophils in the bloodstream [52]. In most cases, this type of NPs is internalized by neutrophils and transported to the tumor site. It should be noted that, in these studies, neutrophils were pre-activated through the administration of TNF-α (tumor necrosis factor-α) or through photosensitization.

It is known that neutrophil adherence to the vessel wall can increase vascular permeability by stimulating the endothelial cells through tumor necrosis factor release and other mechanisms [53]. Thus, even without direct neutrophil involvement in the formation of clusters or microleakages, these types of extravasation could be associated with these cells.

The circulation time also differed between the NPs studied. The MNPs-Cy5 and especially the PLGA-Cy5 NPs circulated for a shorter time than the liposomes-DiD. Rapid clearance by the reticuloendothelial system is a major challenge for polymeric NPs, as it leads to a poor efficiency of NPs’ targeted delivery to the tumor site [54]. The internalization of these NPs by neutrophils can prolong their circulation time and thus increase the delivery efficiency to the tumor.

In conclusion, we clearly showed that the contribution of neutrophils to NPs’ tumor delivery depends on the specific type of NPs. Depending on these properties, some types of NPs are mainly delivered to tumors by neutrophils. Here, neutrophils can directly transport NPs to the tumor tissue or promote NPs’ extravasation in the tumor microenvironment. Other NPs use different tumor accumulation pathways (e.g., spontaneous macroleakage). Therefore, determining the specific characteristics of NPs to facilitate their interactions with neutrophils and taking into account the biological properties of these cells will allow for the creation of a very promising cell-based platform to increase the efficiency of anticancer nanodrug therapy.

## 4. Materials and Methods

### 4.1. Nanoparticles Synthesis

#### 4.1.1. MNPs-Cy5

Magnetic nanoparticles were obtained via the thermal decomposition of iron (III) acetylacetonate (10.48 g) in benzyl alcohol (220 mL). For the MNPs’ (suspension of 80 mg in a 30 mM aqueous NaOH solution) coating with protein, an aqueous solution of human serum albumin (HSA) (8 mg/mL, 20 mL) was used. For coating the obtained NPs with polyethylene glycol (PEG), 5 mg of MNPs-HSA was dissolved in PBS buffer (pH 7.4) (to an Fe^3+^ concentration of 0.95 mg/mL). Then, 266 µL of N-hydroxysuccinimide (NHS) solution in PBS (10 mg/mL) and 457 µL of 1-ethyl-3-(3-dimethylaminopropyl)carbodiimide (EDC) solution in PBS (10 mg/mL) were added. The reaction mixture was incubated with stirring for 10 min, after which, 470 μL of polyethylene glycol hydrochloride salt NH_2_-PEG-OH solution in dH_2_O (10 mg/mL) was added. The incubation time was 1 h. The resulting MNPs-HSA-PEG were separated from the PEG excess via gel filtration using a NAP-10 column (Sephadex G25, eluent PBS, GE Healthcare Bio-Sciences, Chicago, IL, USA). For the MNPs-HSA conjugation with the fluorescent dye Cyanine 5 amine (Cyanine5, an amine derivative, Lumiprobe, Hanover, Germany), 126 µL of EDC solution in PBS (10 mg/mL) and 189 µL of NHS solution in PBS (10 mg/mL) were added to 9 mL of the MNPs (3.5 mg Fe^3+^/mL). The reaction mixture was incubated for 15 min at room temperature. After that, 315 μL of Cy5 solution (1 mg/mL in DMSO) was added. Incubation with a fluorescent label was carried out for 15 h at room temperature. The excess label was removed using a mini-column PD-10 (Sephadex G25, eluent-PBS).

#### 4.1.2. PLGA-Cy5

For the PLGA-Cy5 NPs’ synthesis, a polymer preliminarily modified with a Cy5 fluorescent label was used. The fluorescent dye Cyanine 5 amine was covalently linked to the carboxyl end group of a lactic-glycolic acid copolymer (PLGA, Resomer^®^ 502H, Mw 7–17 kDa, η = 0.21; Evonik Röhm GmbH, Weiterstadt, Germany) by the means of the carbodiimide method in 2 steps. Weighed polymer (1.939 g), EDC (6.6 mg [42.8 µmol]), NHS (2.5 mg), and diisopropylamine (DIEA) (22 mL) were dissolved in methylene chloride. The reaction mixture was stirred overnight at room temperature in the dark. Then, a dye solution (Cy5 2.8 mg (4.28 µmol)) in methylene chloride and, additionally, EDC (1.7 mg (8.5 µmol)) were added. The reaction was carried out for 48 h at room temperature in the dark. The reaction mixture was washed three times with a mixture of equal volumes of water and methanol to remove water-soluble by-products and unreacted starting reagents. The organic phase was separated in a separating funnel, dried over anhydrous sodium sulfate, and evaporated on a rotary evaporator. The resulting precipitate was dissolved in ethyl acetate (approximately 10 mL of ethyl acetate per 1 g of polymer) and added to a tenfold volume of hexane to precipitate the polymer, filtered, and dried in a desiccator. The formation of the conjugate and the absence of impurities were determined using thin layer chromatography (TLC) on plates (eluent methylene chloride:methanol:water 6.5:2.5:0.4, *v*/*v*). The content of Cy5 in the conjugate was measured spectrophotometrically (λ_abs_ = 630 nm) Shimadzu UV-1800, and fluorescence spectra were obtained using a Shimadzu RF-6000 spectrofluorimeter. Fluorescently labeled NPs were obtained through the method of simple emulsions (oil/water). Weighed portions of PLGA polymers (300 mg of the original and 300 mg of the fluorescently modified one) were dissolved in 12 mL of methylene chloride. The resulting solution was added to 60 mL of a 1% solution of polyvinyl alcohol (9–10 kDa) in distilled water and homogenized on an UltraTurrax T18 mechanical disperser with a G10 nozzle for 2 min at 23,600 RMP with cooling; then, on a high-pressure homogenizer Microfluidizer^®^ M -110P (1.5 min at 15,000 psi), residual organic solvent was removed under vacuum. Later, the nanosuspension was filtered, a cryoprotectant (2.5% D-mannitol) was added, poured into vials (1.25 mL each), and lyophilized (Alpha 2–4 LSCplus, Martin Christ GmbH, Osterode am Harz, Germany). Freeze-dried samples were stored at 4 °C.

#### 4.1.3. Liposomes-DiD

The liposomes were obtained via thin lipid film hydration. Egg lecithin (10 mg) (PanReac AppliChem, Darmstadt, Germany), DSPE-PEG2000 (3.4 mg) (Avanti Polar Lipids, Alabaster, AL, USA), cholesterol (3.4 mg) (Avanti Polar Lipids, USA), and DiD (1,1′-dioctadecyl-3,3,3′,3′- tetramethylindodicarbocyanine, 4-chlorobenzenesulfonate salt) fluorescent dye (15 μg) (Invitrogen, Waltham, MA, USA) were dissolved in 1 mL of chloroform at the bottom of a round bottom flask. To form a thin lipid film, chloroform was removed on a rotary evaporator (45 °C, 120 rpm, 200 Pa). The formed lipid film was hydrated at room temperature by adding 1 mL of 10 mM sodium phosphate buffer (PBS, pH 7.4). The lipids were dispersed by stirring for 15 min and then by sonication in an ultrasonic bath for 1 min (22 kHz). To obtain a homogeneous dispersion of liposomes, the solution was sequentially extruded through carbon membranes with pore sizes of 0.4, 0.2, and 0.1 µm, respectively. For the experimental work in vitro and in vivo, fluorescent liposomes were prepared with a final lipid concentration of 16.8 mg/mL in 10 mM PBS. The concentration of DiD in the composition of the prepared liposomes solution was determined using spectrophotometry at an absorption wavelength (λ_abs_) equal to 644 nm (ε_DiD_ = 244,000 L/cm/mol).

### 4.2. Nanoparticles Characterization

#### 4.2.1. Transmission Electron Microscopy

The investigation of the NPs’ morphology was carried out using a transmission electron microscope (TEM) JEOL JEM-1400 (120 kV). For this purpose, 3 μL of MNPs-Cy5 aqueous solution was placed on a copper mesh coated with a formvar film until it dried completely. To study the PLGA-Cy5 NPs, 3 μL of NPs solution in PBS (pH 7.4) was placed on a carbon-coated mesh until completely dry, and negative contrasting was performed: a drop of 1% uranyl acetate aqueous solution was applied to the mesh surface 3 times for 10 sec and removed with filter paper. For liposomes-DiD, 3 μL of NPs solution in PBS was mixed with 3 μL of 5% glutaraldehyde and placed on a carbon-coated mesh until completely dry. Then, the mesh with the sample was washed with distilled water 3 times for 20 s, and negative contrasting was performed according to the scheme similar to that described above.

#### 4.2.2. DLS and ζ Potential

The hydrodynamic size of the NPs was measured using dynamic light scattering (DLS). The size of the NPs and their ζ potential were determined using a Zetasizer Nano ZS (Malvern Instruments). Mean values with confidence intervals were obtained from three measurements per sample.

### 4.3. Animals and Tumor Model

All the animal experiments were approved by the N.I. Pirogov Russian National Research Medical University bioethical committee (protocol # 16/2021). Seven- to nine-week-old female BALB/c mice with weights of 20–22 g obtained from Andreevka Animal Center (Andreevka, Russia) were used for the experiments. 4T1 or 4T1-GFP tumors were established by a subcutaneous (s.c.) injection (right hind flank) of 1 × 10^6^ cells. 4T1 (mouse breast cancer cells) were purchased from the American Type Culture CollectionF (ATCC, Manassas, VA, USA). A GFP-expressing tumor cell line was obtained via lentivirus transduction and had previously been characterized for its in vitro and in vivo growth kinetics [55]. The cells were cultured in RPMI-1640 medium (gibco, New York, NY, USA) with 10% fetal bovine serum (FBS, gibco) and 2 mM L-glutamine (gibco).

### 4.4. Flow Cytometry

Blood was collected through a cardiac puncture into syringes containing 2% ethylenediaminetetraacetic acid (EDTA, Fluka Analytical, Morris Plains, NJ, USA), and tumors were harvested from euthanized animals, digested with 0.1 mg/mL of DNAse I and 1 mg/mL of collagenase I, and passed through a 70 nμm nylon mesh. The animals were anesthetized with an intraperitoneal injection of zoletil (50 mg/kg) and xylazine (5 mg/kg) solution. Red blood cells were lysed using Ammonium-Chloride-Potassium (ACK) solution (gibco). The cells were washed 3 times in cold FC buffer (HBSS (gibco), 2% FBS, and 5 mM EDTA), blocked with anti-CD16/CD32 antibodies (dilution 1:100, Clone 93, Biolegend, San Diego, CA, USA) in FC buffer for 30 min at 4 °C, and stained with fluorophore-conjugated antibodies (Ly6G BV421 (Clone 1A8), CD8a BV421 (Clone 53-6.7), CD19 BV421 (Clone 6D5), Ly6C Alexa-488 (Clone HK1.4), CD4 Alexa-488 (Clone GK1.5), CD45 FITC (Clone 30-F11), CD11b PE (Clone M1/70), CD3 PE (Clone 17A2), and B220 PE (Clone RA3-6B2) in various combinations) or IgG isotype controls (all dilutions 1:100, all from Biolegend, USA), also for 30 min at 4 °C. To detect the level of ROS production, the cells were additionally stained with the fluorescent dye 2′,7′-dichlorodihydrofluorescein diacetate (H2DCFDA, ThermoFisher Scientific, Waltham, MA, USA). In this case, HBSS was used instead of the FC buffer. The cells were incubated in the presence of 10 µM H2DCFDA (in HBSS) for 30 min at 37 °C. Subsequently, the cells were washed 3 times in FC buffer and analyzed on a MoFlo flow cytometer (Beckman Coulter; Miami, FL, USA).

To identify various subpopulations of blood leukocytes interacting with the NPs, as well as to detect the level of ROS production and CD11b expression in the neutrophils, 4T1-bearing mice were intravenously (i.v.) injected with NPs (in PBS) at the following concentrations: (1) 5 mg/kg MNPs-Cy5 (by iron) and 19.5 μg/kg (by Cy5); (2) 50 mg/kg PLGA-Cy5 NPs (by PLGA) and 36 μg/kg (by Cy5); and (3) 70 mg/kg liposomes-DiD (by lipids) and 57 μg/kg (by DiD).

Each experimental group contained 3 animals. Mice i.v. injected with PBS were used as a control group.

Additionally, to study the production of ROS and the expression of CD11b by the activated neutrophils, cells isolated from the blood of the tumor-bearing mice at the 6th day were incubated with phorbol myristate acetate (PMA, Sigma, New York, NY, USA) at a concentration of 100 nM for 30 min in RPMI-1640 medium. Later, they were washed with HBSS and stained as described above.

Summit V5.2.0.7477 software was used for data processing. The leukocyte subpopulation percentages were calculated relative to CD45^+^ cells. In the study of ROS production and CD11b expression, fluorescence intensity values of the H2DCFDA dye and CD11b PE antibody were obtained for the neutrophils that interacted with the NPs and those that did not. Fluorescence from the Cy5 (for MNPs and PLGA NPs) or DiD (for liposomes) dyes in the group that did not interact with the NPs (“Neutrophils without NPs”) was equal to that for the control samples.

### 4.5. Neutrophils Isolation from the Blood of Tumor-Bearing Mice

The blood of the tumor-bearing mice was collected at the 6th day after the 4T1 cell implantation through cardiac puncture into syringes containing 2% EDTA. Plasma was separated from the obtained blood via centrifugation at 400× *g* over 20 min at room temperature. To remove erythrocytes from the cellular component of the blood, it was mixed with 6% Ficoll 400 (Serva-Feinbiochemica GmbH & Co, Heidelberg, Germany) solution in 0.9% NaCl (1:2, *v*:*v*). The suspension was left for 1 h at room temperature. Later, the cellular component was twice washed in PBS/0.2% EDTA solution. The isolation of the pure neutrophil fraction was carried out with the gradient method using Percoll Plus (Cytiva GE Healthcare’s), prepared according to the manufacturer’s instructions. The cellular component was loaded onto a Percoll 60%/80% gradient and centrifuged at 1500× *g* for 45 min at room temperature. The neutrophil fraction was collected at the boundary between Percoll 60% and 80%, washed twice in PBS/0.2% EDTA solution, and resuspended in RPMI-1640 culture medium. Cell counting and a viability analysis were performed on a TC20 Automated Cell Counter (Bio Rad, Hercules, CA, USA) using trypan blue dye. Fraction purity was assessed with flow cytometry after staining the cells with Ly6G BV421, CD45 FITC, and CD11b PE antibodies. As a result, the isolated neutrophils viability was 97%, the population purity—(98.2 ± 0.75)%.

### 4.6. Dynamics of Nanoparticles Interaction with Neutrophils In Vitro

Neutrophils isolated from the blood of the tumor-bearing mice were seeded on coverslips in 48-well plates (ibidi) at a concentration of 5 × 10^5^ cells per well in a culture medium containing blood plasma (1:1, *v*:*v*). Then, the investigated NPs were added to the cells and cultivated for 10–120 min at 37 °C. The final concentrations of NPs in the growth medium were: 100 μg/mL (by iron) and 0.4 μg/mL (by Cy5) for MNPs-Cy5, 1 mg/mL (by PLGA) and 0.72 μg/mL (by Cy5) for PLGA-Cy5, and 1.68 mg/mL (by lipids) and 1.1 μg/mL (by DiD) for liposomes-DiD. Cells incubated in culture medium without NPs or in the presence of free Cy5 dye (in corresponding concentrations of 0.4 μg/mL and 0.72 μg/mL) were taken as controls. Later, the samples were washed 3 times with PBS, stained with fluorophore-conjugated antibodies Ly6G BV421 (Clone 1A8, dilution 1:100, Biolegend, USA) for 30 min, washed 3 times with PBS, fixed with 4% formalin in PBS over 15 min, placed in Dako Mounting Medium (Agilent Technologies, Santa Clara, CA, USA), and studied with a confocal microscope Nikon Eclipse Ti-E A1R MP (Plan Apo 20×/0.75 Dic N objective (numerical aperture = 0.75); Nikon, Japan). The accumulation of NPs in the neutrophils was analyzed by calculating the average intensity of the fluorescence signal from the NPs in cells using the NIS-Elements AR software V.5.15. First, binary masks were created on the Ly6G signal to identify neutrophils in the image. Large masks created on groups of neutrophils were manually segmented. Then, the binary masks were converted into separate regions of interest (ROIs), and the mean fluorescence intensities of the NPs’ signal were measured under each ROI. The fluorescence intensity in the control cells was taken as zero. During the analysis, 130 neutrophils were counted for each NP type.

### 4.7. LDH Assay

The isolated neutrophils were seeded in a 96-well plate (Corning, Somerville, MA, USA) at a concentration of 150,000 per well in triplicate for each NP type and concentration. The cells were left for 1 h at 37 °C and 5% CO_2_ to allow them to attach. Then, NPs were added to the cells and incubated for 2 h. Cells cultured in the growth medium (Spontaneous LDH activity) and in the presence of a lysis buffer (Maximum LDH activity) included in the commercial LDH Cytotoxicity Assay Kit (ThermoFisher Scientific) were used as controls. The assay for assessing the release of LDH was carried out in accordance with the manufacturer’s instructions. After the incubation time, 50 µL of the medium from the cells was transferred into the wells of a new 96-well plate. In total, 50 µL of the reaction mixture from the commercial kit was added to the same wells, pipetted, and left for 30 min at room temperature in the darkn. Finally, 50 μL of stop solution was added and the absorbance of the obtained samples was analyzed on a plate analyzer (EnSpire 2300 Multilabel Reader, PerkinElmer, Waltham, MA, USA) at 490 and 680 nm. The percentage of NP cytotoxicity was calculated using the following formula:$$\%\;Cytotoxicity=\frac{100*\left(«Compound\;treated\;LDH\;activity»-«Spontaneous\;LDH\;activity»\right)}{\left(«Maximum\;LDH\;activity»-«Spontaneous\;LDH\;activity»\right)}$$

Survival was recalculated as (100%-% cytotoxicity).

### 4.8. Intravital Microscopy (IVM)

Animals with a subcutaneously implanted tumor were anesthetized on the 6–7th day, and the tumor was prepared for in vivo observation as described in [56]. Briefly, the tail vein was cannulated using a polyethylene tube (0.28 × 0.60 mm, InStech Laboratories, Plymouth Meeting, PA, USA), and fluorescent antibodies (combinations of 3 μL of Ly6G BV421, 5 μL of Ly6C Alexa-488, and 3 μL of CD11b or 7 μL of CD31 BV421, 5 μL of Ly6C Alexa-488, and 3 μL of CD11b, all from Biolegend) were injected to stain the host cells. The tumor was carefully separated from the underlying tissues and the animal was placed on the heated stage of an inverted confocal microscope Nikon Eclipse Ti-E A1R MP microscope. IVM was performed using a Plan Apo 20×/0.75 Dic N objective (numerical aperture = 0.75; Nikon, Tokyo, Japan) and Apo LWD 40×/1.15 S water immersion objective (numerical aperture = 1.15; Nikon, Japan). Images were scanned sequentially using 405, 488, 561, and 647 nm diode lasers in combination with a DM405/488/561/633 nm dichroic beam splitter. Seven to ten spots were imaged at the time and within 1 h after i.v. injection of 100 μL of NPs (5 mg/kg of MNPs-Cy5 (by iron) and 19.5 μg/kg (by Cy5), 50 mg/kg of PLGA-Cy5 (by PLGA) and 36 μg/kg (by Cy5), or 70 mg/kg of liposomes-DiD (by lipids) and 57 μg/kg (by DiD)) with an acquisition rate of 2–3 frames/min. Individual images were also taken over 100 min. To compare the behaviors of different NP types in the tumor vessels, they were injected successively, wherein MNPs-Cy3 were used (5 mg/kg (by iron) and 12.5 μg/kg (by Cy3)). Five to ten mice were studied for each NP type. To calculate the NPs’ circulation time, fluorescence time-lapse values were recorded in multiple vessels over 50 min after particle administration and averaged. The obtained videos and images were processed in the NIS-Elements AR software V.5.15.

### 4.9. Immunohistochemical Staining (IHC)

Animals with a subcutaneously implanted tumor were anesthetized 1 h after PLGA-Cy5 NP i.v. injection. The tumors were cut out and fixed in a 4% formaldehyde solution over night at +4 °C. Later, they were washed 3 times for 20 min in PBS and placed in TissueTek O.C.T. Compound gel for freezing at −80 °C. Then, the tumors were cut using a freezing microtome HM525 (Thermofisher Scientific). The obtained slices were permeabilized in a 0.1% Triton X-100 solution for 1 h at room temperature, washed, incubated in a 1% solution of goat serum for 1 h at room temperature, and placed in a solution of primary antibodies to NG2—Rabbit Anti-NG2 Chondroitin Sulfate Proteoglycan (Merck, Rahway, NJ, USA)—and smooth muscle actin—Mouse Anti-Actin, α-Smooth Muscle—Cy3™ antibody (Merck)—for 1 h at room temperature. After this, the slices were carefully washed and placed in a solution of secondary antibodies—Goat anti-Rabbit IgG (H+L) Highly Cross-Adsorbed Secondary Antibody, Alexa Fluor 488 (Thermofisher Scientific)—for 1 h at room temperature. The washed stained samples were mounted using ProLong™ Gold Antifade Mountant with DAPI (Invitrogen) and studied with a confocal microscope Nikon Eclipse Ti-E A1R MP.

### 4.10. Whole Mount Staining

Animals with a subcutaneously implanted tumor were anesthetized 1.5 h after liposomes-DiD i.v. injection. The tumors were cut out and fixed in a 4% formaldehyde solution for 4 h at room temperature. After washing, the samples were permeabilized in a 0.1% Triton X-100 solution for 2 h at room temperature, washed, incubated in a 5% solution of goat serum for 2 h at room temperature, and placed in a solution of primary antibodies to NG2 and smooth muscle actin for 24 h at room temperature. Later, the tumors were carefully washed and placed in a solution of secondary antibodies for 24 h at room temperature. The obtained samples were studied with a confocal microscope Nikon Eclipse Ti-E A1R MP.

### 4.11. Statistics

The statistical analysis was performed in GraphPad Prism 9. The data distribution was analyzed using a Shapiro–Wilk test. To compare the normally distributed samples, we used an unpaired *t*-test and one-way ANOVA followed by Tukey’s multiple comparison test. *p* values < 0.05 were considered to be significant.

## Figures and Tables

**Figure 1 pharmaceuticals-16-01564-f001:**
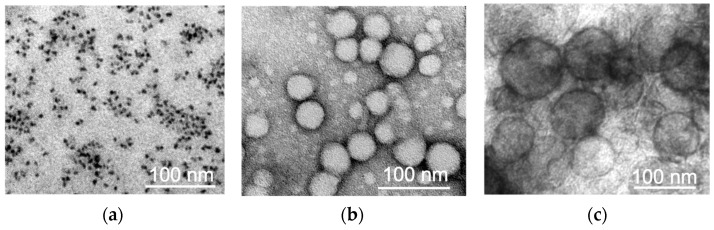
Transmission electron microscopy micrographs of three different types of nanoparticles: (**a**) MNPs-Cy5; (**b**) PLGA-Cy5 NPs; and (**c**) liposomes-DiD.

**Figure 2 pharmaceuticals-16-01564-f002:**
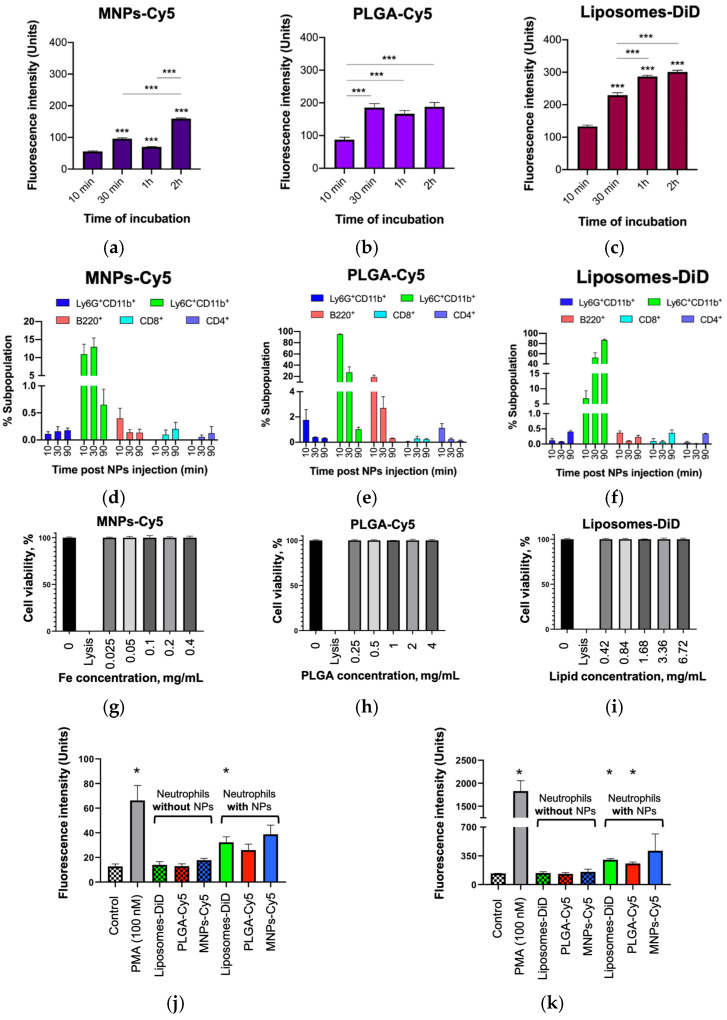
Dynamics of nanoparticle interaction with neutrophils and their effect on cells: (**a**–**c**) NPs’ in vitro interaction with neutrophils isolated from the blood of 4T1-bearing mice at day 6 after tumor implantation, confocal microscopy; (**d**–**f**) NPs in vivo interaction with leukocyte subpopulations after i.v. injection to 4T1-bearing mice at day 6 after tumor implantation, flow cytometry. Results are presented as mean ± SEM. *** *p* < 0.001 (one-way ANOVA followed by Tukey’s multiple comparison test). Subpopulations: Ly6G^+^CD11b^+^—neutrophils, Ly6C^+^CD11b^+^—monocytes, B220^+^—B-lymphocytes, CD8^+^—T_k_-lymphocytes, and CD4^+^—T_h_-lymphocytes; (**g**–**i**) NPs’ toxicity to neutrophils isolated from the blood of 4T1-bearing mice at day 6 after tumor implantation, LDH assay, results are presented as mean ± SD; (**j**,**k**) ROS production (**j**) and CD11b expression (**k**) by neutrophils 30 min after NPs i.v. injection, flow cytometry, results are presented as mean ± SEM, * *p* < 0.05 compared to control group (*t*-test). Following NP concentrations were used for in vitro study (values are indicated per mL of growth medium): 100 µg/mL (iron) and 0.4 μg/mL (Cy5) for MNP-Cy5, 1 mg/mL (PLGA) and 0.72 μg/mL (Cy5) for PLGA-Cy5 NPs, and 1.68 mg/mL (lipids) and 1.1 μg/mL (DiD) for liposomes-DiD; for in vivo study: 5 mg/kg (iron) and 19.5 μg/kg (Cy5) for MNP-Cy5, 50 mg/kg (PLGA) and 36 μg/kg (Cy5) for PLGA-Cy5 NPs, and 70 mg/kg (lipids) and 57 μg/kg (DiD) for liposomes-DiD.

**Figure 3 pharmaceuticals-16-01564-f003:**
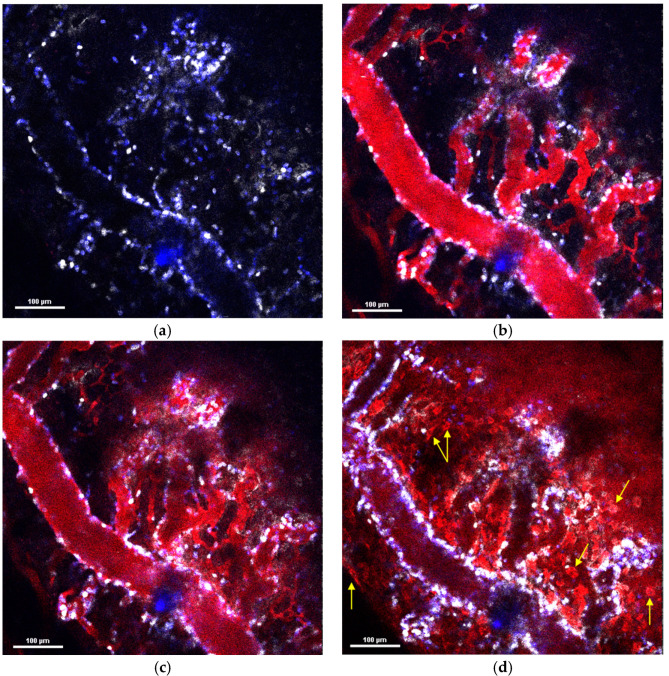
Routes of MNPs-Cy5 extravasation after i.v. administration to 4T1 tumor-bearing mice, IVM: (**a**–**d**) dynamics of NPs’ behavior in tumor vessels and their extravasation due to spontaneous makroleakage—(**a**) micrograph of the tumor microenvironment before MNPs-Cy5 injection, (**b**) the moment of NPs administration, (**c**) 1 min after injection, and (**d**) 8 min after injection. Yellow arrows show the cells of tumor microenvironment that accumulated MNPs-Cy5 (examples). Ly6G^+^—blue, CD11b^+^—white, and MNPs-Cy5—red.

**Figure 4 pharmaceuticals-16-01564-f004:**
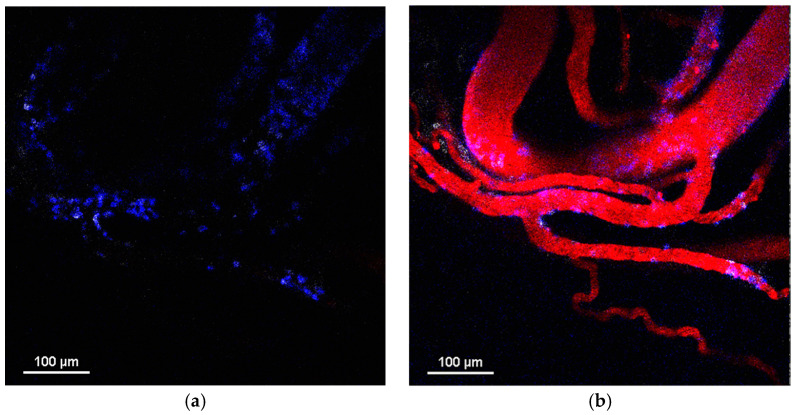
PLGA-Cy5 extravasation after i.v. administration to 4T1 tumor-bearing mice, IVM: (**a**–**d**) dynamics of NPs’ behavior in tumor vessels and their accumulation along the vessel periphery in the form of local clusters—(**a**) micrograph of the tumor microenvironment before PLGA-Cy5 injection, (**b**) the moment of NPs’ administration, (**c**) 5 min after injection, and (**d**) 20 min after injection, arrows show the NP clusters along the vessel periphery; (**e**) PLGA-Cy5 interaction with neutrophils (Ly6G^+^) inside the blood vessel 50 min after injection; and (**f**) neutrophil-based PLGA-Cy5 transport in tumor microenvironment 1 h after NP administration, arrows show the neutrophils with NPs conglomerates, dotted lines represent the vessel wall. Ly6G^+^—blue, CD11b^+^—white, and PLGA-Cy5—red.

**Figure 5 pharmaceuticals-16-01564-f005:**
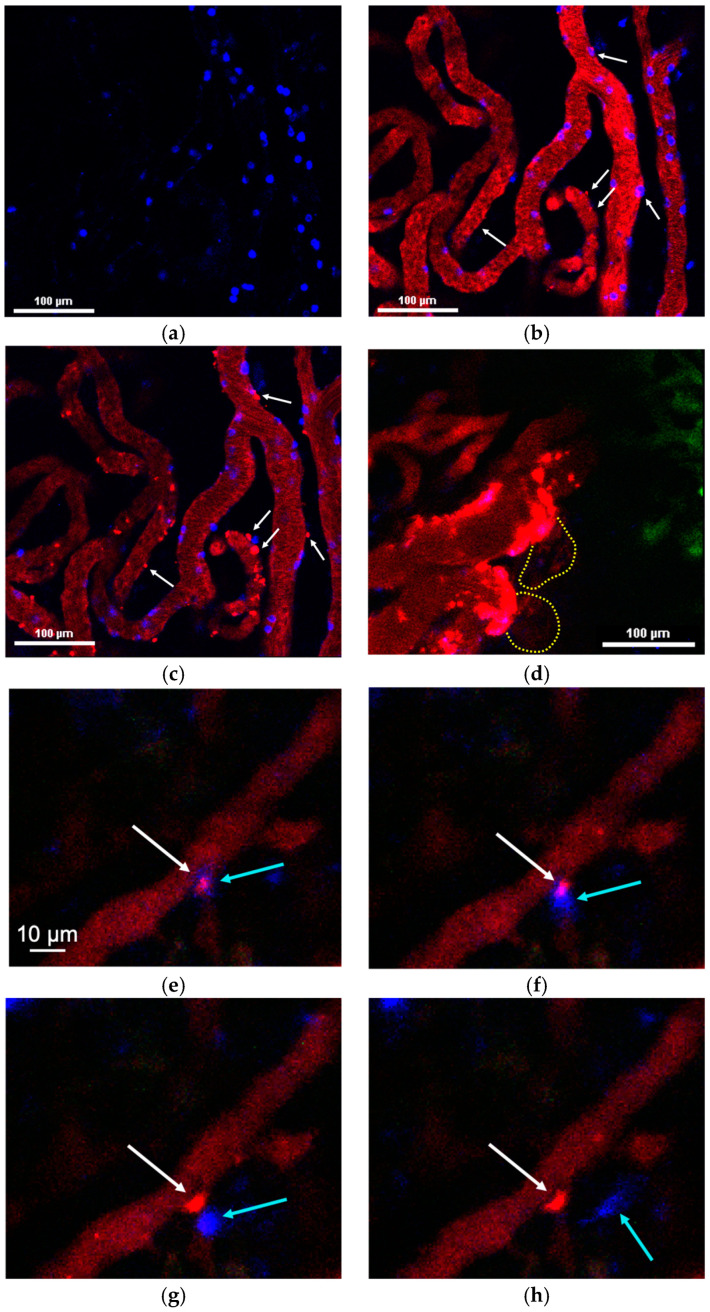
The ways for liposomes-DiD extravasation after i.v. administration to 4T1 tumor-bearing mice, IVM: (**a**–**c**) dynamics of NPs’ behavior in tumor vessels and their extravasation due to microleakage—(**a**) micrograph of the tumor vessel before liposomes-DiD injection, (**b**) the moment of NPs administration, and (**c**) 20 min after injection, arrows show the microleakages formation; (**d**) macroleakages formation, the dotted lines represent the boundaries of the leakages; and (**e**–**h**) microleakage formation mediated by neutrophil extravasation, blue arrow shows neutrophil, white arrow—microleakage. Ly6G^+^—blue, 4T1-GFP—green, and liposomes-DiD—red.

**Figure 6 pharmaceuticals-16-01564-f006:**
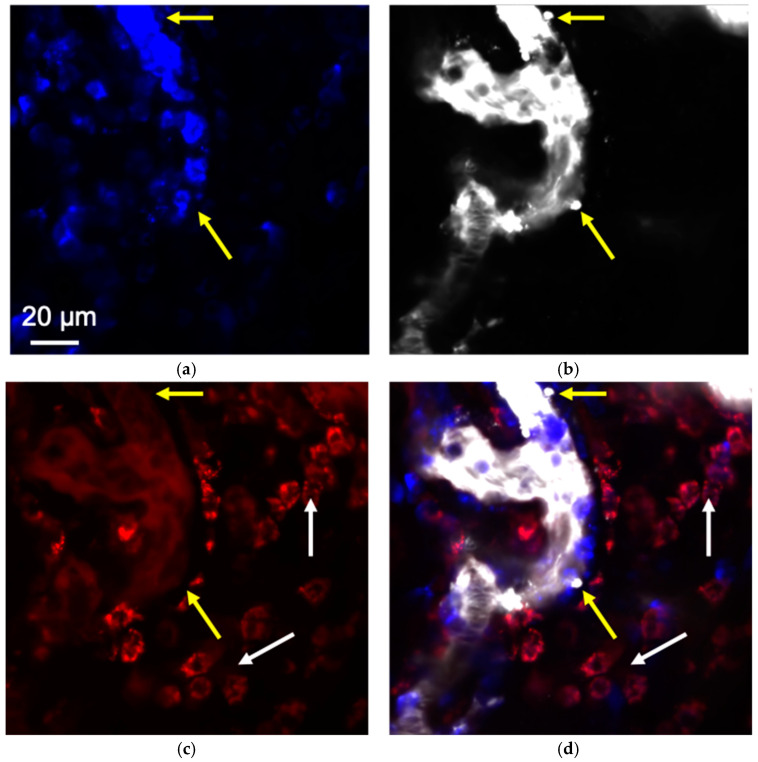
Liposomes-DiD and MNPs-Cy3 extravasation from the tumor vessel after simultaneous i.v. administration to 4T1 tumor-bearing mice, IVM: (**a**) micrograph of tumor vessel and microenvironment in the channel for Ly6G detection; (**b**) micrograph of tumor vessel and microenvironment in the channel for DiD detection; (**c**) micrograph of tumor vessel and microenvironment in the channel for Cy3 detection; (**d**) micrograph of tumor vessel and microenvironment after channels’ merge. Yellow arrows indicate liposomes-DiD microleakages, white arrows—the cells of tumor microenvironment with accumulated MNPs-Cy3 inside. Ly6G^+^—blue, MNPs-Cy3—red, and liposomes-DiD—white.

## Data Availability

Data is contained within the article or Appendix A.

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
