# Peer review of "Neutrophil as a Carrier for Cancer Nanotherapeutics: A Comparative Study of Liposome, PLGA, and Magnetic Nanoparticles Delivery to Tumors"

_pharmaceuticals, 2023, doi:10.3390/ph16111564_

Round 1

Reviewer 1 Report

Comments and Suggestions for Authors

The manuscript by Garanina et al, entitled “Neutrophil as a carrier for cancer nanotherapeutics: a comparative study of liposome, PLGA and magnetic nanoparticles delivery to tumors” covers an important issue focusing on the features of neutrophils-NPs interactions to analyze the ability of neutrophils to act as carriers for nanomaterials-based therapeutic agents. The manuscript is well structured. However, some issues should be clarified:

Major issues:

1.       Were the nanoparticles stable? Zeta potential values, especially for liposomes, indicate that the stability is questioned.

2.       Introduction lacks data about the currently established biological effects of the studied nanomaterials.

3.       Staining protocol for H2DCFDA staining is not provided.

4.       Why the selected concentrations of nanomaterials were used?

5.       Why parametric tests were used? How the distribution normality was assessed?

6.       Figure 2a-c contain no indications of statistical significance (e.g., asterisks). Nor these data are mentioned in the text. In Figure 2d-f, replace commas with dots.

7.       Figure S1. Why pan-leukocyte marker CD45 was used to assess the content of leukocytes in blood? Or did the authors assess the percentage of neutrophils among CD45+ cells? It is not clear based on the data in Figure. The same issue can be addressed concerning Figures 3S and 4S.

8.       Provide representative histograms of DCF fluorescence and CD11b  expression in Figure 2.

9.       ROS overproduction is a common marker of nanoparticle toxicity. Thus, its observation in neutrophils in response to the nanoparticles studied may be indicative of their toxicity. This should be discussed in the manuscript.

1-  Limitations are not mentioned.

Minor issues:

- - Abstract. Nanomaterials slightly increased ROS production and CD11b expression. Add whether the difference was statistically significant.

1-  IgG isotype controls (all from Biolegend). Mention the country of manufacturer

--  MoFlo flow cytometer (Beckman Coulter). Mention the country of manufacturer

1-   Figure 2s. Legend should be expanded to include channels used.

1-   Figure 3. Please use arrows to show NPs accumulation.

Comments on the Quality of English Language

Must be improved. The meaning of some sentences is difficult to grasp. 

Author Response

The authors thank the Reviewer for careful reading of the manuscript and positive response. We are grateful for Reviewer’s valuable suggestions and considered all points to achieve a stronger paper. Below, we give the point-by-point reply to the Reviewer’s remarks and suggestions.

Reviewer 2 Report

Comments and Suggestions for Authors

The article titled " Neutrophil as a carrier for cancer nanotherapeutics: a compara-2 tive study of liposome, PLGA and magnetic nanoparticles de-3 livery to tumors  presents a comparative study on the use of neutrophils as carriers for cancer nanotherapeutics, specifically liposomes, PLGA, and magnetic nanoparticles. The study investigates the interaction of these nanoparticles with neutrophils both in vitro and in vivo, their behavior in blood vessels, and their ability to deliver the nanoparticles to tumors. Overall, the article provides valuable insights into the potential of neutrophil-mediated delivery systems for cancer treatment.

Structure and Clarity:

The article is well-structured and follows a logical flow. The introduction provides background information on nanomedicine and the challenges associated with nanodrug accumulation in tumors. The authors effectively highlight the significance of the study. The methods section describes the experimental procedures in sufficient detail, allowing for reproducibility. The results section presents the findings clearly, supported by appropriate data. However, the conclusion section is relatively brief and could be expanded to provide a more comprehensive summary of the study's outcomes.

Methodology:

The methods employed in the study are adequately described. The authors utilized confocal microscopy, flow cytometry analysis, and intravital microscopy techniques to investigate the interaction between neutrophils and different types of nanoparticles. However, additional details regarding the specific parameters and settings used in these techniques would be valuable for better reproducibility. Furthermore, the authors should clarify the sample sizes used in the in vitro and in vivo experiments and provide information on statistical analyses performed.

Results and Data Analysis:

The results presented in the article are informative and supported by experimental data. The authors successfully demonstrated that neutrophils efficiently interacted with liposomes, PLGA, and magnetic nanoparticles, leading to enhanced drug delivery to tumors. The in vivo observations using intravital microscopy revealed the accumulation of neutrophil-loaded nanoparticles in tumor vessels. However, the authors should provide more quantitative data, such as the percentage of neutrophils containing nanoparticles and the level of tumor accumulation, to support their conclusions. Additionally, statistical analyses should be performed to determine the significance of the observed differences between the nanoparticle types.

Interpretation and Discussion:

The authors provide a reasonable interpretation of the results and discuss the implications of their findings. They correctly emphasize the potential of neutrophil-mediated delivery systems for improving cancer treatment. However, the discussion could be further strengthened by comparing the results with existing literature and addressing any discrepancies or limitations. Furthermore, the authors should discuss the clinical relevance and future directions of their research.

Ethical Considerations:

The article does not mention any ethical considerations, such as animal welfare or human subjects. It is important to address whether the study followed ethical guidelines and obtained appropriate approvals.

Significance and Originality:

The study addresses an important topic in nanomedicine and contributes to the understanding of neutrophil-mediated delivery systems for cancer treatment. The comparative approach involving liposomes, PLGA, and magnetic nanoparticles adds originality to the research. However, it would be valuable to discuss the novelty of the study more explicitly and provide a more comprehensive analysis of the significance of the findings.

Questions for Clarification:

-          What was the rationale for selecting liposomes, PLGA, and magnetic nanoparticles for the comparative study? Were there any specific advantages or properties of these nanoparticles that made them suitable for neutrophil-mediated delivery?

-          Were there any specific criteria used to choose the peripheral blood neutrophils as carriers for the nanoparticles? What is the relevance of using peripheral blood neutrophils in the context of cancer treatment?

-          Can you provide more details about the techniques used for confocal microscopy, flow cytometry analysis, and intravital microscopy? Specifically, what parameters were measured, and how were the images analyzed?

-          How were the nanoparticles loaded into the neutrophils? Was there any specific loading strategy employed, and how was the efficiency of loading determined?

-          What were the sample sizes used in the in vitro and in vivo experiments, and were they statistically determined? Did the authors consider any potential variability or bias in the experimental design?

-          What statistical analyses were performed to assess the significance of the results? Were the observed differences between the nanoparticletypes statistically significant? Please provide the specific statistical tests used and the reported p-values.

-          Did the study encounter any limitations or challenges during the experimental procedures, and how were these addressed? Were there any specific controls or validation experiments performed to address potential confounding factors?

-          Were there any observed differences in the behavior of neutrophils when interacting with different types of nanoparticles, and were these differences statistically significant? Please provide quantitative data or statistical values to support the conclusions.

-          How do the findings of this study compare to existing research on neutrophil-mediated drug delivery systems? Were these comparisons statistically analyzed? Please discuss any discrepancies or similarities in the results.

-          What are the potential implications of this research for the development of targeted cancer nanotherapeutics? Were these implications discussed in light of statistical significance? How does this study contribute to advancing the field?

Th article provides valuable insights into the use of neutrophils as carriers for cancer nanotherapeutics. The study's findings support the potential of neutrophil-mediated delivery systems for enhancing nanodrug accumulation in tumors. However, to strengthen the article, it is recommended to provide more detailed methodology descriptions, perform robust statistical analyses, address ethical considerations, and further discuss the novelty, significance, and clinical relevance of the study. Additionally, the language and presentation should be carefully reviewed and improved. Furthermore, providing more quantitative data and discussing the results with statistical values would enhance the analysis and interpretation of the findings.

I recommend that the article undergo significant revisions addressing the specific comments mentioned. The revisions should include more detailed methodology descriptions, robust statistical analyses, addressing ethical considerations, and further discussing the novelty, significance, and clinical relevance of the study. Once the revisions are completed, the article can be considered for publication.

Comments on the Quality of English Language

The overall language and presentation of the article are clear and concise. However, there are a few instances where sentence structure or grammar could be improved. A thorough proofreading for language and typographical errors is recommended.

Author Response

(The authors gave the same response as above.)

Reviewer 3 Report

Comments and Suggestions for Authors

.           This paper describes in vitro and in vivo study of 3 different types of nanoparticles with various sizes and surface charge properties for the delivery of therapeutics via nanodrug cell-based delivery systems. The authors reference previous studies and use a mouse model that shows an increased levels of circulating neutrophils as their model and target cell to load with nanoparticles. Authors used intravital microscopy to demonstrate clearance of particles from circulation, and also used this technique to study the dynamics of particle accumulation within the tissues which was very interesting as they could show quite clear differences between the 3 particle types and how they interacted with neutrophils. The authors also used electron microscopy to demonstrate the physical differences in size and properties of the nanoparticles. The paper is generally written well, but has a few grammatical errors and some parts could do with some more information.

Specific comments below:

.           Line 58 typo "... while 70 nm particles were nor not captured..." 

.           Reword lines 61 to 63 for clarity.

.           Line 69 "Another Other NPs characteristics,..."

.          Line 82 "the tumor growth decrease and the an increase in animals’ survival

.           Please use "taken-up" instead of uptaken in lines 87, 114, 203 & 331 uptaken is not a word

.           Please change inflammation to inflammatory in line 93

.           Line 94 "... the endowing NPs with..." please change to "functionalisation of NPs with ..."

.           Which experiments are the authors referring to? Please be specific Line 180

.           Please change "Nanoparticles'" to "Nanoparticles" in lines 121, 135, 198, 367, 440 & 516 

.           Please consider changing the title of this section "2.1. Nanoparticles’ characteristic" line 121 to "Nanoparticle characterisation"

.           What fluorescent dyes are used? Be specific. line 131

.           Line 138-139 "... blood isolated from the tumor bearing mice blood, revealed..."

.           Grammar line 141: "Figure 2. Dynamics of nanoparticles interaction..."

.           Line 144: "particles i.v. injection to 4T1-bearing..."

.           Please be more clear in line 151 regarding the dosage of NPs. Is this 100 μg/mL (iron) for MNP-Cy5, 1 mg/mL (PLGA) for PLGA-Cy5 NPs, and 151 1.68 mg/mL (lipids) for liposomes-DiD/ml is this per ml of blood? 

.           Line 161 change "...1.5 hours of following NPs injection."

.           Line 174 change "... perspective carriers..." to "preferred carriers..."

.           “NPs types caused increased cell death even...” line 179. General comment for this statement: I find it hard to believe that there is no cell death that occurs during the process of collecting and extracting neutrophils from these mice that there appears to be no death in the cells. Has this data been normalised to non-treated animals as a control so that there is a background of cell death? Please state this in the analysis. 

.           Which experiments are the authors referring to? Please be specific Line 180

.           Line 181: "Another factor determining the perspective suitability of using NPs..."

.           Line 182 grammar: "...effect on cells activation."

.           Line 200 change to: "... and the ways routes of extravasation"

.           Line 207 change to: "Figure 3. The ways Routes of MNPs-Cy5 extravasation"

.           Line 209 typo: "makroleakages" change to "macroleakage"

.           Line 213 grammar:  "PLGA-Cy5 NPs extravasated by accumulating..."

.           Line 215 grammar: "Besides, NPs were found..."

.           Line 220 grammar: "...PLGA-Cy5 clusters could be pinpointed, elongated..."

.           Line 237 please simplify sentence. It’s hard to understand.

.           Line 241, please explain the sentence "In case of liposomes-DiD, the pattern of their behavior in blood vessels was different." Different to what?

.           Line 260 please change to "To confirm the difference in the ways routes of NPs extravasation..."

.           Line 268 please change to "... clearance from the vessels occurred during 40 min after administration, ..."

.           Line 269 suggested sentence change from "The most long circulated NPs were liposomes-DiD." to:  "NPs with the longest blood circulation time were liposomes-DiD" maybe add in this information how long did these particles persist in the blood for?

.           Line 293: "In this current study, three types..."

.           Line 300: Please elaborate on accumulationof NPs in cells… Do the authors mean accumulation in tumour cells? Or do they mean accumulation in immune cells?

.           Line 302 Change to "5) the ways routes of extravasation..."

.           Line 305 Grammar: "... absence of neutrophils toxicity [14]." 

.           Line 308 Grammar: "... for cells loading [36]."

.           Line 310 Grammar: "The same data were was obtained by Bisso et al. [14]"

.           Line 316-317 change to “In first the case of liposomes-DiD accumulated..."

.           Line 322 “… an increase in cells ROS production 7 CD11b” Do the authors mean neutrophil cells or immune cells in general?

.           Line 342-343 Change to: "... demonstrate the same behaviour as PLGA NPs behavior in vivo [41].”

.           Line 343 “...characteristic extravasation way method for NPs..."

.           Line 348 Grammar: "It is known that neutrophils adherence..." 

.           Line 353 “...PLGA-Cy5 NPs circulated for less time than liposomes-DiD."

.           Line 354 “… is a significant challenge of polymeric NPs[43]” challenge for what? Challenge for delivery at the tumour site? Or something else? I do not understand this sentence. Please be specific Please elaborate

.           Line 357 Grammar: "demonstrated that neutrophils contribution to..."

.           Line 364 grammar: “... will allow to create the creation of a very promising cell-based…” 

Comments on the Quality of English Language

The English language quality could be improved as some parts are difficult to understand. I have made suggestions for improvements to the language in the previous section.

Author Response

(The authors gave the same response as above.)

Round 2

Reviewer 1 Report

Comments and Suggestions for Authors

The authors have provided responses and amendments. However, some issues still remain unclear.

1.       Aggregation of nanomaterials can modulate toxicity. The authors claim that the used nanoparticles are stable for several days. Is there any data on the accumulation of NPs in tissues and features of their excretion from the body?

2.       Data on the assessment of distribution normality should be added to the manuscript.

3.       Representative histograms on ROS signaling should be added to the manuscript. According to the histograms, the number of cells varied significantly. How can it be explained? Was the viability of cells assessed during H2DCFDA staining? DCF fluorescence should be assessed in viable cells only to avoid data misinterpretation.  

Comments on the Quality of English Language

It is desirable to show the manuscript to a native speaker 

Author Response

We thank the Reviewer for overall positive response. We have added additional information to the paper to address the comments. Below, we give the point-by-point reply to the Reviewer’s questions.

Reviewer 2 Report

Comments and Suggestions for Authors

Accepted

Author Response

We thank the Reviewer for the positive response and the suggestion to accept this manuscript.

Round 3

Reviewer 1 Report

Comments and Suggestions for Authors

The comments have been addressed. 

Comments on the Quality of English Language

Can be improved.